

# Determination of the affinity constants for phage display albumin-binding peptides

Yi-Feng Shi

Department of Biotechnology, School of Biological Engineering, Dalian Polytechnic University, Dalian, Liaoning Province, P. R. China

## ABSTRACT

**Background**. Phage display technology has been established as a powerful screening approach to select ligands or peptides for binding to proteins. Despite rapid growth in the field, there has been a relative dearth of quantitative criteria to measure the effectiveness of the process of phage display screening. Since human serum albumin (HSA) has been extensively studied as a drug carrier to extend the plasma half-life of protein therapeutics, the use of phage display technology is required for identifying albumin-binding peptides as the very promising strategy of albumin-binding against albumin fusion. The construction of albumin-binding drug requires the assessment of a large quantity of HSA-binding peptide (HSA binder) candidates for conjugation with therapeutic proteins. The use of the linear epitope mapping method has allowed researchers to discover many HSA-binding peptides. However, it may be inefficient to select these peptides based on sequence identity via randomly sequencing individual phage clones from enrichment pools.

**Method**. Here, a simple assessment method to facilitate phage display selection of HSA-binding peptides was recommended. With experimentally determined phage titer, one can calculate the specificity ratios, the recovery yields and the relative dissociation constants, which are defined as quantitative criteria for panning and characterization of the binding phage fused peptides.

**Results**. Consequently, this approach may not only enable more rapid and low-cost phage display screening, but also efficiently reduce pseudo-positive phages selected as HSA binders for conjugation with therapeutic proteins.

Corresponding author
Yi-Feng Shi, shiyf@dlpu.edu.cn

## INTRODUCTION

Human serum albumin (HSA) is a 67 KDa globular protein with a concentration of 3.4 to 5.4 g/dL in blood, accounting for about 60% of the total protein in blood serum. Its long half-life, 19~20 days, makes HSA an ideal carrier to extend the half-lives of therapeutic proteins (*Varshney et al., 2010*; *Dennis et al., 2002*; *Larsen et al., 2016*; *Cho et al., 2022*). Both HSA fusion and HSA binding technologies have been developed to reduce the clearance rate of peptide and protein drugs. HSA fusion technologies, e.g. granulocyte colony-stimulating factor-HSA (Albugranin) (*Halpern et al., 2002*), growth hormone-HSA

(Albutropin) (*Osborn et al., 2002a*; *Osborn et al., 2002b*), interleukin 2-HSA (Albuleukin) (*Melder et al., 2005*), interferon-HSA (Albuferon- $\alpha$) (*Osborn et al., 2002a*; *Osborn et al., 2002b*), glucagon-like peptide-1 receptor agonists-HSA (*Yu et al., 2018*) and clotting factor IX-HSA (*Pasca & Zanon, 2022*) have been achieved through expression of HSA-fused proteins via gene recombination. In contrast, HSA-binding technologies attach an HSA binding peptide or protein domain rather than HSA itself to therapeutic proteins, enabling them to bind to HSA through non-covalent interactions (*Hoogenboezem & Duvall, 2018*). For example, AB-Fab is trastuzumab-Fab fused to HSA binding peptide (AB) (*Nguyen et al., 2006*), scDb-ABD is the albumin-binding domain of streptococcal protein G (ABDs) fused to a bispecific single chain diabody (*Stork, Müller & Kontermann, 2007*) and AlbudAb/IL-1ra is interleukin-1 receptor antagonist fused to serum albumin-binding domain antibodies (AlbudAbs) (*Holt et al., 2008*).

Which HSA technology is right for extending *in vivo* life of a protein drug? *Walker et al. (2010)* compared such two forms of recombinant IFN- $\alpha$2b: serum albumin-binding domain antibodies fused to IFN- $\alpha$2b (AlbudAbs- IFN- $\alpha$2b) and HSA directly fused to IFN- $\alpha$2b (HAS-IFN- $\alpha$2b). They found that AlbudAbs-IFN- $\alpha$2b showed 18-fold higher potency in cell reporter assays, 1.5-fold increased half-life in rat PK studies and 5.8-fold greater antiviral efficacy in the EMCV assays (Encephalomyocarditis virus assay for antiviral activity) (*Walker et al., 2010*). *Shi & Bian (2008)* compared a series of protein drugs created by fusion (covalent) and binding (non-covalent) to HSA and found that the HSA-fusion proteins showed a more significant loss in binding capacity to their receptors, which related directly to their pharmacological action. The decreased pharmaceutical effects of HSA-fusion drugs were different than what was observed for HSA-binding proteins because the HSA molecule is much larger than HSA binders resulting in steric hindrance and structural heterogeneity. In addition, the HSA-binding peptides are smaller in size and do not greatly increase the size of recombinant proteins, thus often present an advantage in their ability to diffuse rapidly into tissues (*Shi & Bian, 2008*).

Due to the binding properties with various ligands, HSA generally transports endogenous and exogenous compounds, which might be toxic in the unbound state, but non-toxic as albumin-bound. Thus HSA may serve as a circulating depot for improving the pharmacokinetic behavior of a variety of drugs, including: drug half-life in the bloodstream, regulating drug efficacy and decreasing drug toxicity, which based on the unbound substances are the pharmacologically active moieties. Since HSA-binding technologies have been shown to efficiently improve the half-life of protein drugs, it is essential to obtain high affinity and specificity of HSA binding peptides or HSA binders. Phage display technology has been successfully applied to antibody library and random peptide library technologies since its emergence in 1985 (*Zhang & Shi, 2010*; *Smith, 1985*; *Kehoe & Kay, 2005*), and provides a convenient platform for screening HSA-binding peptides with the required properties. The use of the linear epitope mapping method has allowed researchers to discover many HSA-binding peptides (*Dennis et al., 2002*). However, it may be inefficient to select these peptides based on sequence identity *via* randomly sequencing individual phage clones from enrichment pools. In this study, the protocols to isolate HSA-binding

peptides based on phage titer and identify the relative affinity of albumin-binding peptides were developed.

Phage display technology has been established as a powerful screening approach to select ligands or peptides for binding to proteins. Despite rapid growth in the field, there has been a relative dearth of quantitative criteria to measure the effectiveness of the process of phage display panning. It is also necessary to measure directly the target-binding-affinity constants of the phage-fused-peptide rather than the artificial binding-peptide synthesized after phage display screening. This study focuses on developing general criteria for the performance of rapid and cost-effective phage display screening.

## MATERIALS AND METHODS

### Phage library and bacteria strains

The Ph.D.-7 Phage Display Library Kit including the Heptapeptide Phage Display Library and *E. coli* ER2738 host strain were purchased from New England Biolabs. ER2738 with F factor, which confers tetracycline resistance, can grow in LB-Tet medium (1% bacto-tryptone, 0.5% yeast extract, 0.5% NaCl, 20 g/mL tetracycline) and is used for M13 phage propagation. Since the library phage carries the *lac* Z gene, phage plaques appear blue when the phage-infected ER2738 cells are plated on LB-Tet-IPTG-X-gal media (1% bacto-tryptone, 0.5% yeast extract, 0.5% NaCl, 20 µg/mL Tetracycline, 50 µg/mL IPTG, 40 µg/ml X-gal, 1% agarose).

### Phage titer analysis

A single *E. coli* ER2738 colony was inoculated into 10 mL LB-Tet medium and incubated at 37 °C by shaking until mid-log phase ($OD_{600}{\sim}0.5$) for phage titering. Phage solutions were diluted 10-fold serially in TBS (50 mM Tris–HCl, pH 7.5, 150 mM NaCl) and aliquots of these solutions (50 µL) were added to a suspension of the prepared ER2738 cells (450 µL), 1 for each phage dilution. The phage-infected *ER2738* cells (100 µL) were then poured from each dilution tube to an LB-Tet-IPTG-X-gal plate and the plates were incubated overnight at 37 °C. We determined the number of phage particles contained in the original stock phage culture by counting the number of plaques formed on the seeded agar plate and multiplying this by the dilution factor.

### Panning procedure

96-well microplates (Polypropylene, Xinhua Glass Instruments Factory, Sanhe, Haimen, China) were coated overnight at 4 °C with 100 µL of the HSA target (Sigma, 100 µg/mL in 0.1 M NaHCO$_3$, pH 8.6) and blank control (0.1 M NaHCO$_3$, pH 8.6) respectively. After rapid washing three times with TBST (TBS + 0.1% [v/v] Tween-20), 100 µL of the diluted phage library ($\sim$2 $\times$ 10$^{11}$ phage) was added to the well of the coated plate and rocked gently to let them absorb for 60 min. Following 10 washes with TBST, the bound phages were eluted with 0.2 M Gly-HCl, pH 2.2. The 100 µL that was eluted was neutralized with 15 µL Tris–HCl (1 M, pH 9.1) immediately. A small amount of eluted phage was titered as described above and the rest of elutes were added to 20 ml ER2738 culture (OD $_{600}{\sim}$0.5) to be amplified 4.5 h at 37 °C for the next round.

### Recovery yield and specificity ratio

To evaluate the efficiency of the phage display panning procedure, we created two selection criteria: the recovery yield and the specificity ratio. The recovery yield is the ratio of the titer of the eluted phage to the input phage. The specificity ratio is the ratio of the titer of the eluted phage from the target coated well to the blank control well.

$$\text{Recovery Yield (\%)} = \frac{\text{Elute phage titer from target}}{\text{Input phage titer}} \times 100$$

$$\text{Specificity Ratio} = \frac{\text{Elute phage titer from target}}{\text{Elutephage titer from blank}}.$$

### Isolation and purification of phage

The amplified culture of phage clones was transferred to a 50 mL centrifuge tube and centrifuged for 10 min at 10,000 rpm at 4 °C with a high-speed refrigerated centrifuge (Himac CR21G). The supernatant was decanted to a fresh tube and precipitated by 1/5 volume of PEG/-NaCl (20% [w/v] Polyethylene Glycol–8000, 2.5 M NaCl) at 4 °C overnight and centrifuged for 20 min at 10,000 rpm. The supernatant was discarded and the sediment containing phage was suspended with TBS. To precipitate the phage again, a 1/5 volume of PEG/NaCl was added to the culture and it was stored at 4 °C for at least 60 min. The PEG suspension was centrifuged for 10 min at 10,000 rpm, the phage precipitate was suspended in 500 µL of TBS.

### Measurement of the relative dissociation constant of selected peptide fused phage binding to target

The dissociation constant ($K_d$) can be determined from a double-reciprocal plot of eluted phage concentrations $[\phi]_e$ against a variety of input phage concentrations $[\phi]_t$ at a fixed target concentration of HSA [T] according to the following equation (see Results & Discussion in detail):

$$\frac{1}{[\phi]_e} = \frac{K_d}{[T][\phi]_t} + \frac{1}{[T]}.$$

The $K_d$ value is equal to the slope of the line $K_d/[T]$ divided by the intercept on the vertical axis 1/[T], in which the target concentration of HSA immobilized in the microplate well is not necessary for a $K_d$ determination.

### DNA sequencing for characterization of specific binding peptides

After 3 rounds of bio-panning, individual phage clone (blue plaque) was isolated, amplified and purified to check the specificity ratio of binding to HSA. The sequences of HSA-binding peptides were determined by DNA sequencing of the selected phage using −96 gIII sequencing primer 5ʹ-HOCCC TCA TAG TTA GCG TAA CG-3ʹ in the Ph.D-7 phage library.

**Table 1** Phage titers during panning against HSA (In 200 μg/mL HSA coated microplate).

| Panning | Round 1 | Round 2 | Round 3 |
|---|---|---|---|
| Input | | | |
|    HSA (pfu/ml) | $4.40 \times 10^{11}$ | $4.24 \times 10^{11}$ | $2.40 \times 10^{11}$ |
|    Blank (pfu/ml) | / | $4.24 \times 10^{11}$ | $2.40 \times 10^{11}$ |
| Elute | | | |
|    HSA (pfu/ml) | $5.20 \times 10^{4}$ | $7.24 \times 10^{4}$ | $1.12 \times 10^{5}$ |
|    Blank (pfu/ml) | / | $3.08 \times 10^{4}$ | $1.44 \times 10^{4}$ |

# RESULTS

## HSA-immobilizing microplates and blocking agents

Unlike the instructions on many phage display peptide library kits, we do not need to coat the well with a blocking agent protein such as BSA, skim milk, *etc.*, which process is common with ELISA assays though. Blocking proteins are a mixture that may cause more complex binding results in phage display screening. If the number of phages binding to blocking proteins is larger than to target proteins, the recovery yield still rises in each round of panning and may cause pseudo-positive results in phage display screening. Thus we used a blank well as a negative control. The specificity ratio is used to measure how much the number of target binding phages is larger than the number of blank binding phages in each round of the panning process. Therefore no matter what kind of microplate, with low or high protein-binding capacity, is used to immobilize the target protein molecule, the specificity ratio is generally suited for the assessment of the presence of target-binding phages.

## Panning against HSA

We examined the enhancement effects of using the titer-based criteria in monitoring panning HSA-binding peptides from the Ph.D.-7 phage display peptide library. The phage titer results in the panning were shown in Table 1 and the corresponding recovery yields and specificity ratios were shown in Table 2. Our data in 200 μg/mL HSA coated microplate demonstrated both a recovery yield and specificity ratio increase by each round panning. Specifically, the recovery yield increased 1.45 fold in the second round and 3.6-fold in the third round, as in the first round, and the specificity ratios improved significantly from 2.35 in the second round up to 7.78 in the third round. Importantly, the specificity ratio in panning rounds 2 and 3 that is larger than 1 indicated that the eluted phages preferred binding to HSA in the target coated well over the blank in the target-uncoated well. The increase in specificity ratio after each round of panning also implies that phages with high affinity for HSA have been amplified. Because blank micro-plates only function to distinguish target-binding phages from microplate–binding phages in panning results, the first round of panning using the original phage display peptide library against the blank microplate is not necessary and should be excluded from the panning process. The increase in the recovery yields after each round indicates that the pools of high affinity phages binding to HSA have been enriched.

**Table 2** The recovery yields and the specificity ratios during panning against HSA (In 200 $\mu$g/mL HSA coated microplate).

| Panning | Round 1 | Round 2 | Round 3 |
|---|---|---|---|
| Recovery yield (%) | $1.18 \times 10^{-7}$ | $1.71 \times 10^{-7}$ | $4.25 \times 10^{-7}$ |
| Specificity ratio | / | 2.35 | 7.78 |

**Notes.**
The recovery yields were calculated by the titers of eluted phage divided by the titers of input phage; the specificity ratios were calculated by the titers of eluted phage from target molecule coated plate divided by the titers of eluted phage from blank control plate.

**Table 3** Specificity ratios, relative dissociation constants and sequences of phage clones selected for binding to HSA.

| Phage displyed peptide | Input phage titre (pfu/ml) | Elute phage titre (pfu/ml) | Blank control phage titre (pfu/ml) | Specificity ratio | Relative dissociation constant ($\mu$mol/L) | DNA sequence | Peptide sequence |
|---|---|---|---|---|---|---|---|
| HSA-3 | $1.16 \times 10^{12}$ | $9.16 \times 10^{5}$ | $1.24 \times 10^{5}$ | 7.4 | 0.529 | GCG GAG AGT TAG AAG CTT AAT | AESQKLN |
| HSA-4 | $1.34 \times 10^{13}$ | $5.12 \times 10^{6}$ | $5.22 \times 10^{5}$ | 9.8 | 0.388 | CTT ACG CCG CTT CCG TTG CGT | ITPLPLR |

## Identification of positive HSA-binding phages

The specificity ratio was used to determine if positive phage clones with high HSA-binding affinity were isolated. From the final round of panning plates, we isolated individual clones to amplify, purify and measure the specificity ratio for binding to HSA. We tested the selected clones with significant specificity ratios as potential HSA-binders, after screening many phage clones that consisted of microplate-surface-binding phages, the specific and non-specific target-binding phages, from the target-binding plate as shown in Table 3. The two selected clones, HSA-3 and HSA-4, show specificity ratios of 7.4 and 9.8, respectively, confirming that our method in the panning process is effective.

## Phage binding assay by the relative dissociation equilibrium constants

For confirming the binding peptides to HSA, the dissociation equilibrium constant ($K_d$), which is commonly used to describe the interaction between a ligand (such as a drug) and a protein, is also employed to explore how tightly the selected peptide which fused with phage binds to the target molecule. The formation of the binding complex ($\phi T$) of phage fused peptide and HSA can be described in a simple one-to-one association of a phage fused peptide ($\phi$) and a target molecule HSA (T)

$$\phi + T \rightleftharpoons \phi T. \tag{1}$$

The corresponding dissociation constant ($K_d$) is defined as:

$$K_d = \frac{[T][\phi]}{[\phi T]} \tag{2}$$

where [T], [$\phi$] and [$\phi$T] represent molar concentrations of the target molecule HSA, phage fused peptide and their binding complex, respectively.

The total target concentration $[T]_t$ is equal to the concentration of bound phage target [$\phi$T] and free target [T]:

$$[T]_t = [T] + [\phi T]. \tag{3}$$

Substituting this expression for [T] in equation of dissociation constant ($K_d$):

$$K_d = \frac{[\phi]([T]_t - [\phi T])}{[\phi T]}. \tag{4}$$

Thus, the bound and total target and the free phage concentration are related by rearranging above equation of dissociation constant ($K_d$):

$$\frac{[\phi T]}{[T]_t} = \frac{[\phi]}{K_d + [\phi]}. \tag{5}$$

The concentration of bound phage [$\phi$T] and free phage [$\phi$] can be calculated from the mass conservation equation, where [$\phi$]$_T$ is the total concentration of input phage and the concentration of target bound phage [$\phi$T] can be estimated by eluted phage concentration [$\phi$]$_e$, which is very much less than [$\phi$]$_T$ in our experiment.

$$[\phi] = [\phi]_t - [\phi]_e \approx [\phi]_t \tag{6}$$

$$[\phi T] \approx [\phi]_e. \tag{7}$$

A double reciprocal plot of phage-target affinity measurement is generated by plotting $1/[\phi]_e$ as a function of $1/[\phi]_t$. The slope is $K_d/[T]$, the intercept on the vertical axis is $1/[T]$, and the intercept on the horizontal axis is $-1/K_d$. Thus, the dissociation constant ($K_d$) can be readily derived from eluted phage concentrations at a variety of total input concentrations at a fixed target concentration of HSA.

$$\frac{1}{[\phi]_e} = \frac{K_d}{[T][\phi]_t} + \frac{1}{[T]}. \tag{8}$$

This method is used to evaluate the relative dissociation constants of selected phage fused peptide HSA-3 and HSA-4 against target molecule HSA respectively. Figures 1, 2 shows a double reciprocal plot for the determination of the relative dissociation constant of selected peptide fused phage HSA-3 and HSA-4 binding to HSA respectively. In the experiment, a constant concentration of target molecule HSA coated in the wells of microtiter plates was incubated with varying the concentrations of input HSA-3 or HSA-4 peptide fused phage until equilibrium was reached. Then TBST buffer wash the unbound phages, followed by the elution of the bound phages to the well with glycine-HCl, pH2.2. The concentration of input and eluted phage was quantitated by titration. The $K_d$ value HSA-3 and HSA-4 were conveniently calculated from linear regression of the plot. The relative order of affinity of the selected peptides HSA-3 and HSA-4 displayed on phage for the target HSA are shown in Table 3. These phage-fused peptides were subjected to DNA sequencing and their corresponding heptapeptide sequences are also shown in Table 3.

## DISCUSSION

### The recovery yield and the specificity ratio

To analyze the effectiveness of our phage display experimental technique, two operating parameters, the recovery yield and the specificity ratio, were developed. These parameters

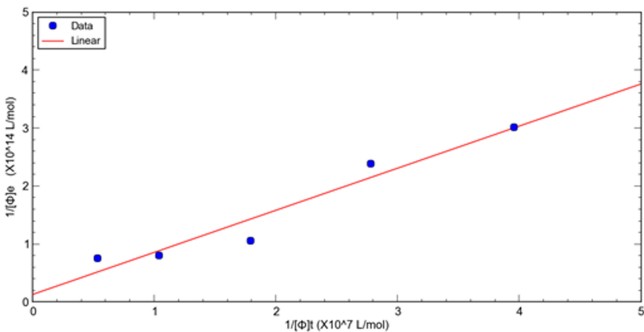

**Figure 1** A double reciprocal plot for relative affinity constant measurement of human serum albumin binding peptide HSA-3 fused phage.

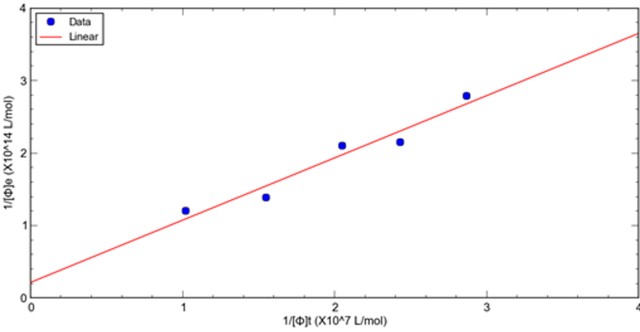

**Figure 2** A double reciprocal plot for relative affinity constant measurement of human serum albumin binding peptide HSA-4 fused phage.

assessed the initial efficiency of phage display screening. The recovery yield was calculated by the titer of eluted phages divided by the titer of input phages in the same target-coated well. The value of recovery yield quantifies the enrichment of target-binding phages. Ideally, the recovery yield will rise as the phage display proceeds and the enrichment of phages after each round of panning progresses so that a greater and greater quantity of target-binding phages is achieved. However, the specificity ratio is more necessary than the recovery yield in monitoring the phage display screening experiment. The specific ratio is employed to identify positive clones of target-binding phages, as compared with the blank control. It is calculated by the titer of eluted phages from the target-coated well divided by the titer of eluted phages from the blank control well. The specificity ratio should be larger than 1, for a positive result indicating that target-binding phages are present. High specificity ratios indicate a large amount of target-binding phages and imply that there may be high affinity of phage clones present.

It is very necessary to use the specificity ratio to avoid background interference generated by non-specific binding (*Vodnik et al., 2011*). This is because eluted phages often include both target-binding and microplate surface-binding phages, resulting in false-positive

panning. Low specificity ratios and recovery yields may result from the presence of a large number of plate-surface-binding phages.

According to the instruction manual for the phage display peptide library kit by New England Biolabs (*New England Biolabs, 2006*), large-scale randomized DNA sequencing in the enrichment pool of eluted phages is required to uncover the binding motif among target-binding peptides. But the surface of HSA for binding drugs may not be limited to the motif-specific area. This method focusing on phage-fused peptide clones with greater binding capacity rather than the enriched HSA-binding motif may be a more rapid and cost-efficient means of determining which HSA-binding peptides are candidates for conjugation with a bioactive protein.

### Surrogate measure of intended HSA-binding affinity

The indirect ELISA methods were also reported to determine the affinity constants of phage-displayed peptides (*Thong et al., 2018*; *Parakasikron et al., 2021*), but our method is direct and short protocol, which saves the phage antibody employed. Although biosensors such as surface plasmon resonance (SPR) or quartz crystal microbalance (QCM) are popular methods for the determination of the affinity of protein-protein interaction, the relative dissociation constants ($k_D$) between phage-displayed peptides and their respective targets *via* phage titer developed by us and others (*Dyson, Germaschewski & Murray, 1995*) still can be employed as a surrogate measure of the affinity of protein-protein interaction, which method may be cost-effective and user-friendly for rapid validation of candidates of a target binder. In general, when the HSA-binding peptides are fused with bioactive proteins, their affinity for HSA would most likely decrease owing to shielding effects and folding problems in their corresponding fusion proteins. The accurate affinity of free-HSA binding peptide binding to HSA, as measured by their SPR or QCM-measured $K_D$, does not necessarily have a guaranteed affinity in HSA binding peptide-drug fusions for HSA. Thus, the relative dissociation constants ($k_D$) or even the specificity ratio can be employed as an initial criterion to select HSA-binding peptides as candidates for fusion with the intended protein drugs, then subjected HSA binding peptide-drug fusions to SPR or QCM to examine or validate the $K_D$ between the fusion protein and HSA.

### CONCLUSIONS

Phage display is a general practical tool for selecting or isolating binding molecules against the desired targets in phage libraries. In the case of targeting the protein with epitope mapping method, conventional bio-panning has limitations on the efficient screening of the functionally relevant peptides. With experimentally determined phage titer instead of sequencing individual phage clones from enrichment pools, some quantitative criteria are characterized for bio-panning and binding affinity of phage display of screening binding peptides in this study.

In summary, both the recovery yield and the specificity ratio are designed to monitor the phage display panning process. The specificity ratio is a critical recognition feature used to discriminate target binding from background interference. The relative dissociation constant of a peptide-fused phage also serves as a surrogate endpoint to demonstrate the

extent of binding affinity and relative specificity of an HSA binder. The use of such criteria facilitates phage panning and avoids pseudo-positive phage clones selected for further test procedures. Therefore, these quantitative parameters based on phage titer are simply amenable to a wide variety of phage display screening applications.

The selected HSA-binding peptides will be attached to bioactive proteins to prolong their half-lives through non-covalent association with HSA *in vivo*. The small size makes HSA-binding peptides ideal fusion partners for recombinant proteins because they avoid decreasing biological activity and maintain the native capacity to penetrate tissues. In addition, the extremely high plasma concentration of HSA confers the fusion proteins more opportunities to bind HSA, even with a relatively low affinity HSA-tag. Thus, small HSA-binding peptides are attractive candidates for the engineering of pharmacokinetic properties of protein therapeutics.

## ACKNOWLEDGEMENTS

Y-F Shi thanks Min Li, Jia-Di Zhang, Yan Lu and Lei Bian for their assistance with the experiment.

### Funding

The author received no funding for this work.

### Competing Interests

The author declares there are no competing interests.

### Author Contributions

- Yi-Feng Shi conceived and designed the experiments, performed the experiments, analyzed the data, prepared figures and/or tables, authored or reviewed drafts of the article, and approved the final draft.

### DNA Deposition

The following information was supplied regarding the deposition of DNA sequences:

The phage display HSA binding peptide sequences are available at GenBank: OQ660434, OQ660435.

### Data Availability

The raw measurements are available in the Supplementary Files.

### Supplemental Information

Supplemental information for this article can be found online at http://dx.doi.org/10.7717/peerj.15078#supplemental-information.

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
