# Peer review of "Determination of the affinity constants for phage display albumin-binding peptides"

_PeerJ, doi:10.7717/peerj.15078_

## Round 0.1 · original submission · Minor Revisions

Reviewers have successfully attended to your work, and apologies for the time it has taken. You can find that the reviewers have found a lot of merit to your contribution, and have offered various comments to improve it. Please kindly address the comments to your very best, and provide adequate details where possible.

In addition, Editor considers the author to please consider the following points, that would help add strength to the work, and help readership.

a) It is great how you have introduced HSA technology, however, HSA-binding technologies are not well emphasized. Please, from line 66, it will be useful to provide more information about what underpins HSA-binding technologies, before expressing their importance.

b) Please, the objective statement, kindly sharpen it, to be more direct, 'described the efforts' seems not so strong. Brainstorm on it please. Also, highlight what would be anticipated as the potential outcome of this study. You will see from my remarks below why this is necessary, to connect with the conclusion.

c) Materials and methods is ok, but can be improved by the following:
- create a new subsection called 'overview of experimental program'. in this subsection, in 3-4 sentences, succinctly describe the major stages taken to conduct the entire study, and please support this with a schematic flow diagram. An idea of the flow can be:
Design of study > Assembly of experimental materials > Phage library and bacteria strains development> Phage titer analysis> Panning procedure> Analysis (Recovery yield and Specificity ratio/Isolation and purification of phage/relative dissociation constant /DNA sequencing)
This is a suggestion, apply your discretion here. The essence of this is to guide readers to follow the steps of this study. The editor believes that author will find aspects of these materials and methods that would require improvement through this.
- any parameter measured or studied, please, kindly indicate why that parameter is being studied, with a reference.

d) Please, kindly separate Results and Discussion, consistent with PeerJ requirement. Extract results succinctly, and relay the results. And in the discussion, please, kindly strengthen the discourse with recent literature, where possible. The editor observes that a lot of the literature is quite old (of course that does not make them irrelevant). Please, try to see if you can find more recent literature, it might help add value to the discourse.

e) Conclusions: Please, the editor believes that it will be very useful to start the conclusion by reiterating what the objective of this work was, why this study is relevant, the proposed anticipated outcome, and then, outline the key tools that were utilised to make this study happen. All this should be the first paragraph, before starting the next paragraph (that is , line 254).

This is a very scholarly study, and I look forward to your revised manuscript.

Reviewer 1 ·

Basic reporting

While the language used in the manuscript is mostly understandable and clear, there are some minor issues regarding sentence construction (line 108, 119, 154, 166-168, 231), interpunction, grammar (e.g.line 80), spelling (e.g. lines 266, Table 1) and text formatting (e.g. lines 87, 100, 256, Figure 1). Most of these cases can be solved by a careful proofreading as they arise from missing or extra characters, common grammar errors and formatting.

While the literature referenced outlines the background and deep need for presented research, it appears to be scarce with just 17 articles cited (2 of which authored or co-authored by the author of reviewed work and 1 being a manual for a research kit used to acquire results, which most likely should not be cited in the first place). Reference list does not follow the format suggested and is not sorted accordingly to guidelines, but this is a but minor nuisance.

The article is structured properly, tables and figures are readable, although riddled with spelling errors (Table 1) and text formatting issues (Figure 1).

The author presents a justified case for the significance of his results.

Experimental design

no comment

Validity of the findings

no comment

Additional comments

Careful proofreading is strongly suggested. Manuscript posing a valid, important point and an interesting novel methodology approach of much weight could use a more robust and up to date references list.

Annotated reviews are not available for download in order to protect the identity of reviewers who chose to remain anonymous.

·

Basic reporting

The paper presents useful information in the fields of biotechnology and biomedical sciences, which could aid cost-effective method of drug formulation or delivery. The introduction/background are satisfactory and the article structure look fine. However, the paper needs English language editing. Unconventional acronyms were used without prior definition. Categorical statement/claims were made without citations. Information that should have been presented in the materials and methods section occurred under the result and discussion section.

Experimental design

The paper is within the scope of the journal. The relevant knowledge gaps were identified, stated and addressed. The methodology is sound and sufficient to permit reproducibility although few improvements were suggested in the comments to author attached.

Validity of the findings

The results appear original but statistics appears to be missing. Conclusions were stated.

Additional comments

Specific comments for the author are as listed below:
1. L27-29: Should be “However, …may not be efficient….” Instead of “However, it may be not efficient…”
2. L31-33: Did the author computed the listed parameters? My guess is no. If so, write “….one can calculate the specific ratios….”
3. L35: Insert “of” between “phage-display screening” and “HAS-binding peptides”
4. L 40: What is KDa? Kilodalton? Although KDa is a standard SI unit, it could be confused for "Kills, Deaths, Assists". So consider defining it at this first use.
5. L46-47: It is not good to use the same word twice or more in a sentence. Consider replacing “through” in line 47 with “via”
6. L47: …attach on or attach as…? It do not see the justification for use of “an” in that sentence
7. L 47-49: Any citation to these claims?
8. Line 53: Walker et al should have a citation number. E.g. Walker et al [10]…
9. L 57: Please define the acronym “EMCV”
10. L58: Shi et al should have a citation number. E.g. Shi et al [11]……
11. L59-60: Since you are talking of proteins, delete “a” and change receptor to “receptors”
12. L60-63: Are these findings that of the author? My guess is no. That is why a citation is required to give credit to the rightful author/authors.
13. L66-67: A citation is required after “protein drugs” to substantiate this claim.
14. L72: use “… may not be….” Instead.
15. L72-73: Please give reason why it may not be efficient. E.g. “….enrichment pools because…….”
16. L73-75: Please rephrase the sentence. It can be “we” when this paper has a single author. You may write “In this study, the protocols for isolating HSA-binding peptides based on phage titer was described, focusing on developing criteria for the performance of rapid and cost-effective phage display screening”
17. L81-82: Beyond the content of the LB-Tet medium, please state the producer of the media and country of origin to aid research reproducibility. E.g. (XYZ®, China)
18. L84: insert ”are” between “cells” and “plated”
19. L84-85: Please state the producer of the media and country of origin to aid research reproducibility.
20. L88: ….37◦ C with intermittent shaking until…..?
21. L89: TBS?
22. L93; Number of phage colonies?
23. L96: Ninety-six-well microplates?
24. L119: ….at least 60 min?
25. .. according to the following equation
26. L137 -151: The statements here appear to be more of M&M than results and discussion. The background info being presented is important but this could have come in the M&M section or even in the introduction. Please present your findings as per the recovery yield and the specificity ratio and discus the biotechnology or biomedical implications, if any.
27. Line 153: Please, delete “of”
28. L154: …. With a blocking agent? Please what is “et al”? Do you mean etc.?
29. L153-155: Please, edit the sentence to make it more meaningful.
30. L158: Thus we used…….?
31. L170: ... in the third round?
32. L197: This method of…?
33. L207-208 and elsewhere in this manuscript: If you are going to number the equation, please use equation editor and number the equations serially. This equations can’t be designated as number one because you are inserted some equations before now.
34. L206: are simple and applicable?
35. L269-271: Please, edit the acknowledgement to make it more meaningful.
36. REFERENCE: Insert the suggested citations and renumber the references as appropriate

---

## Round 0.2 · Major Revisions

Author, thank you for your patience, as your manuscript underwent review. Please, sorry that it has taken a while, it has been very challenging to secure reviewers. A reviewer has considered your work and raised very important questions. Please, kindly carefully and diligently attend to them, and provide adequate responses as well.
Looking forward to your revised manuscript. Thank you.

Reviewer 3 ·

Basic reporting

The manuscript presents interesting area of phage display binding peptides, however, some revisions are required

1.. Your abstract needs to be modified. As this is a research manuscript, it is ideal that your abstract starts with a statement on the overall objective/aim of the study. In line 25, correct the grammatical error … “it may be not efficient…” You need to re-write your abstract. The current form does not reflect your work.

2. Your manuscript requires minor English editing and correction of many grammatical errors, which can prevent potential readers from understanding the information you want to pass across. I suggest you use English editing software, or consult English editing service provider, or give it to a native English speaker to help you do the editing

3. Your references are outdated (mostly within 2000 to 2010) and do not show you considered the recent developments in phage display binding peptides before undertaking and reporting your study. You have to consult recent research publications to know the recent development in similar areas and reflect it in your work. I suggest you considered at least five similar studies published within 2016 to date. Some of these reference materials include https://doi.org/10.1016/j.jbiotec.2016.05.027, doi: 10.1007/978-1-4939-9853-1_2, https://doi.org/10.1111/febs.13674, https://doi.org/10.1016/j.jbc.2021.101342, etc.

4. You have to include the objectives of your study in the introduction. You also need to add a little detail, 2 to 4 sentences, on justifying your objective or stating its significance.

Experimental design

5. Your methods need some revision. You need to improve the reporting style. E.g., in line 131, you wrote the sentence as if it’s a proposal. Kindly revise accordingly and apply it throughout the manuscript. You need to provide a little more detail in your methodology. For instance, under “DNA sequencing for characterization of specific binding peptides”, you need to provide a brief information on how you isolated, amplified, and purified the individual phage clone, as well as the names of the equipment used. Kindly refer to this recent publication https://doi.org/10.1038/s41598-019-42628-6 for more insight on a few things related to your work.

Validity of the findings

6. Your results and discussion need improvement. You need to present your arguments and back them up with recent references.

Additional comments

7. You need to add at least one recommendation for future studies in your conclusion

---

## Round 0.3 · accepted · Accept

After a very careful and thorough evaluation of this revised manuscript, I am convinced that this work is acceptable for publication. The author carefully addressed all concerns raised.

Thank you for finding PeerJ as your journal of choice, and look forward to your future scholarly contributions.

Congratulations